# Water-Free SbO$_x$ ALD Process for Coating Bi$_2$Te$_3$ Particles

Sebastian Lehmann [1] , Fanny Mitzscherling [1], Shiyang He [1,2], Jun Yang [1,2], Martin Hantusch [1] ,
Kornelius Nielsch [1,2] and Amin Bahrami [1,*]

1 Institute for Metallic Materials, Leibniz Institute of Solid State and Materials Science, 01069 Dresden, Germany
2 Institute of Materials Science, Technische Universität Dresden, 01062 Dresden, Germany
* Correspondence: a.bahrami@ifw-dresden.de

**Abstract:** We developed a water-free atomic layer deposition (ALD) process to homogeneously deposit SbO$_x$ using SbCl$_5$ and Sb-Ethoxide as precursors, and report it here for the first time. The coating is applied on Bi$_2$Te$_3$ particles synthesized via the solvothermal route to enhance the thermoelectric properties (i.e., Seebeck coefficient, thermal and electrical conductivity) via interface engineering. The amorphous character of the coating was shown by the missing reflexes on the X-ray diffractograms (XRD). A shift from the oxidation state +III to +V of the Sb species was observed using X-ray photoelectron spectroscopy (XPS), indicating increased thickness of the SbO$_x$ coating layer. Additionally, a peak shift of the Sb 3d$_{5/2}$ + O 1s peak indicated increased n-type doping of the material. Electrical measurements of spark plasma-sintered bulk samples confirmed the doping effect on the basis of decreased specific resistivity with increasing SbO$_x$ layer thickness. The Seebeck coefficient was improved for the coated sample with 500 cycles of SbO$_x$, while the total thermal conductivity was reduced, resulting in enhancement of the $zT$. The results distinctly show that surface engineering via powder ALD is an effective tool for improving key properties of thermoelectric materials like electrical conductivity and the Seebeck coefficient.

**Keywords:** powder atomic layer deposition; water-free SbO$_x$ ALD process; Bi$_2$Te$_3$ particles; solvothermal synthesis; thermoelectric materials





## 1. Introduction

Thermoelectric (TE) technology can be considered a viable long-term solution to the energy crisis and environmental pollution. Thermoelectric devices based on Bi$_2$Te$_3$ have been the most commonly used to date for the transfer of waste heat into electrical energy. Since its discovery in the 1960s, their efficiency has been continuously improved by applying several methods, like nanostructuring or surface modification [1]. The efficiency of thermoelectric materials is described by the dimensionless figure of merit ($zT$) value (Equation (1)).

$$zT = \frac{S^2\sigma}{\kappa_{tot}}T \tag{1}$$

where $S$, $\sigma$, $\kappa_{tot}$, and $T$ are the Seebeck coefficient, electrical conductivity, total thermal conductivity, and absolute temperature, respectively. The product $S^2\sigma$ is defined as the power factor. The total thermal conductivity $\kappa_{tot}$ consists of two constituents, lattice and electrical thermal conductivity, $\kappa_{lat}$ and $\kappa_{el}$, respectively. The goal of materials optimization is to establish a balance between these property indicators, especially the decoupling of electrical and thermal properties [1,2]. Massive efforts have been directed towards figuring out how to separate these parameters in order to get around their limitations [3–5]. The interface modification of TE materials with the aim of reducing thermal conductivity and increasing $S$, resulting in great success in improving $zT$ values, is one promising and extensively investigated method. In general, whereas uniformly coating the TE materials with the second phase creates continuous interfaces, adding the second phase to the TE

matrix directly can result in discontinuous interfaces [2]. One tool that can be used to enhance thermoelectric properties via surface or interface modification is Powder Atomic Layer Deposition (Powder ALD). ALD is a well-known deposition method that enables controlled layer-by-layer growth of different materials on planar substrates, as well as three-dimensional objects. Powder ALD has already been used to effectively enhance the TE properties of ZrNiSn-based materials [6], $CoSb_3$ [7], and $Bi_2Te_3$ [8–11] alloys. Total thermal conductivity was suppressed due to the second phase formation and the aforementioned reasons. The deposition of a thin oxide shell around the (TE) particles can efficiently reduce thermal conductivity while insignificantly reducing electrical conductivity [12,13]. Due to the formation of such a phase barrier and the considerable mismatch in acoustic impedance or phonon spectra between the (oxide) shell and the matrix phase, phonons are scattered significantly more than with typical grain boundaries [14].

Many ALD recipes for the deposition of oxides involve conventional oxidizers like water, $O_2$, $O_3$, and $H_2O_2$ [15–17]. For air- and/or moisture-sensitive powders, the use of conventional oxidizers can cause the formation of an uncontrolled oxide layer, which contradicts the basic goal of ALD, which is to precisely control the film thickness. Since $Bi_2Te_3$ degrades in the presence of moisture [18], we developed a water-free process for the deposition of $SbO_x$ using $SbCl_5$ and Sb-Ethoxide as the oxidizer. $SbO_x$ is a well-known dielectric material that can be used to improve the Seebeck coefficient [19]. However, as mentioned before, all ALD-$SbO_x$ processes rely on the use of conventional oxidizers like $H_2O_2$ and $H_2O$ or $O_3$, which can deteriorate the thermoelectric performance of $Bi_2Te_3$ (nano)particles. Furthermore, it is postulated that the amorphous nature of the $SbO_x$ layer causes no lattice mismatches between the substrate and the coating, further reducing the thermal conductivity of the $Bi_2Te_3$. The sintered and compacted particles can, in the end, be used to build a thermoelectric module consisting of leg pairs (n- and p-type) [20].

## 2. Materials and Methods

The $SbO_x$ thin films were deposited onto silicon substrates using a thermal ALD reactor (Veeco Savannah S200) at different deposition temperatures of 80, 90, 100, and 120 °C. Sb-Ethoxide ($Sb(OEt)_3$) and $SbCl_5$ were used as precursors. High-purity $N_2$ was used as the carrier gas, and the chamber was kept at a flow rate of 10 sccm. The optimized pulse/reaction/purge times for one ALD deposition cycle were 0.5/30/40 s (Sb-Ethoxide) and 0.5/30/40 s ($SbCl_5$). The growth rate was 0.48 Å at a deposition temperature of 100 °C, which was determined by utilizing X-ray reflectometry (cf. Figure 1) and confirmed by atomic force microscopy (Figure S1, Supplementary Materials). The sample deposited at 100 °C was subjected to annealing at 350 °C for 20 min to study the thermal stability of the film.

Hexagonal-shaped $Bi_2Te_3$ particles were synthesized via a solvothermal route reported elsewhere [21]. For the synthesis, a milky solution of 1 mmol bismuth chloride (min. 99%, Strem, Newburyport, MA, USA), 1.5 mmol sodium tellurite (99.95%, Alfa Aesar, Kandel, Germany), 8 mmol sodium hydroxide (Merck, Darmstadt, Germany), and 0.5 g polyvinylpyrrolidone MW 40,000 (Alfa Aesar) was poured into a Teflon autoclave. The materials were dissolved in 40 mL ethylene glycol (99+%, Alfa Aesar) beforehand. The preparation of the solution and the filling and sealing of the autoclave were carried out in a glove box under an argon atmosphere. Then, the sealed autoclave was placed in a drying oven (VWR Venti Line 112 Prime) at room temperature, heated to 180 °C at a heating rate of 6 K/min, and held for a holding time of 36 h. The resulting silvery-black solution was centrifuged six times with isopropanol (100%, VWR International, Rosny-sous-Bois-cedex, France) and subsequently dried in a vacuum. Initially, the supernatant solution was dark yellow and streaky, but it became colorless and transparent in the end.

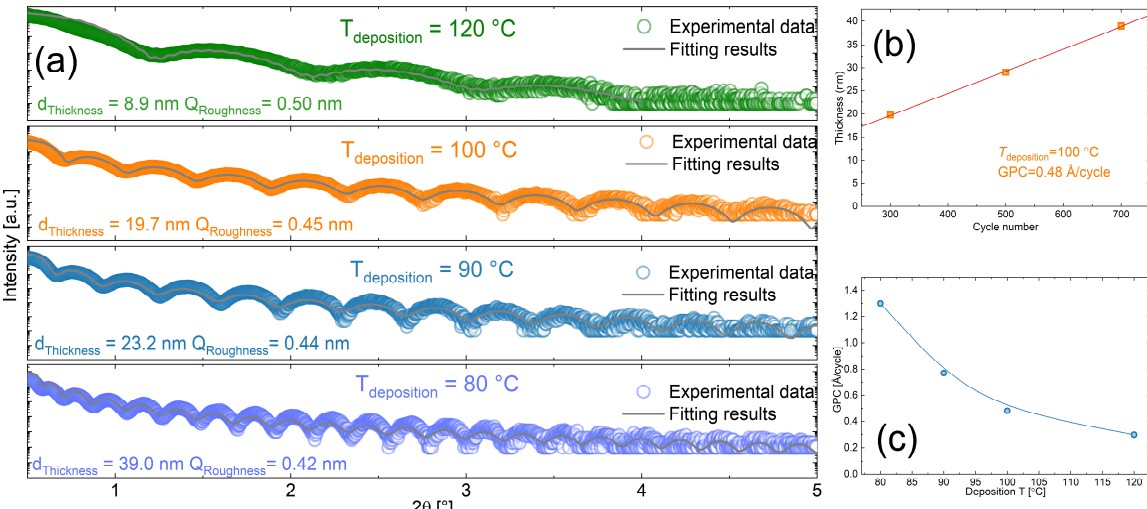

**Figure 1.** (**a**) XRR data of ALD coating with SbO$_x$ thin film for 300 cycles at different deposition temperatures. (**b**) Thickness data determined via XRR plotted vs. cycle number. (**c**) Growth per cycle (GPC) data determined via XRR plotted vs. deposition temperature.

The SbO$_x$ coating around the Bi$_2$Te$_3$ particles was realized at a deposition temperature of 100 °C using a thermal ALD reactor (Veeco Savannah S200) with a rotating titanium tube (rotation speed of 20 rpm), where the Bi$_2$Te$_3$ particles were loaded inside. Note that the temperature inside the tube was presumed to be lower than 100 °C because of the thermal gradient inside the ALD chamber, which should result in a higher growth rate (cf. XRR data (Figure 1)). Sb-Ethoxide and SbCl$_5$ were used as precursors. High-purity N$_2$ was used as the carrier gas, and the chamber was kept at a flow rate of 20 sccm during the ALD reaction process. The optimized pulse/reaction/purge times for one ALD deposition cycle 0.5/30/40 s (Sb-Ethoxide) and 0.5/30/40 s (SbCl$_5$). Upscaling of this coating procedure to industrial scale is easily possible using a larger rotating drum or utilizing a so-called fluidized bed reactor (FBR). With a larger surface area, the exposure time should then be increased.

The pellets were sintered via a spark plasma sintering procedure under vacuum using an SPS 210-Gx (AGUS) device. The coated powder was prepared in a standard graphite tool for the sintering process. Additionally, tungsten foil was added on the bottom and the top between the tool and the powder to avoid sticking of the sample. First, a gradual preheating step up to 300 °C with a pressure of 2.5 MPa (0.2 kN) was carried out to remove any potentially remaining organic impurities. After cooling to 60 °C, sintering was performed at 350 °C with a pressure of 11.5 MPa (0.9 kN) for 5 min. The pellet diameter was 5 mm. The synthesis procedure is shown in the flow chart in Figure 2.

The crystallinity and composition of the uncoated and SbO$_x$-coated Bi$_2$Te$_3$ particles were characterized using X-ray diffraction (XRD, with Co K$\alpha$ radiation, Stadi P diffractometer, STOE, Darmstadt, Germany). The crystallinity and composition of the SbO$_x$ thin film and the sintered Bi$_2$Te$_3$ pellets were characterized using X-ray diffraction (XRD, with Co K$\alpha$ radiation, Bruker, D8 advance, Karlsruhe, Germany). The thickness of the thin-film surface morphology was determined using X-ray reflectometry (XRR, X'Pert MRD PRO) and confirmed by atomic force microscopy (AFM, Dimension Icon, Bruker Karlsruhe, Germany). The morphology of the uncoated and SbO$_x$-coated Bi$_2$Te$_3$ particles, as well as the cross-section images of the sintered pellets, were analyzed using field emission scanning electron microscopy (FE-SEM, Sigma300-ZEISS FESEM, Oberkochen, Germany). X-ray photoelectron spectroscopy (XPS) of the uncoated and coated powders was performed with a PHI 5600 spectrometer using monochromatic AlK$\alpha$ radiation (200 W) with a pass energy of 29.35 eV. A charge correction is not necessary (C 1s at 284.8 eV). The electrical resistivity of the sintered pellets was investigated using a Quantum Design PPMS device. The Seebeck

coefficient was measured at room temperature by LSR-3 (Linseis, Selb, Germany) under an argon atmosphere. The thermal conductivity ($\kappa$) was measured using the laser flash method under a helium atmosphere (LFA 1000, Linseis, Selb, Germany) at room temperature.

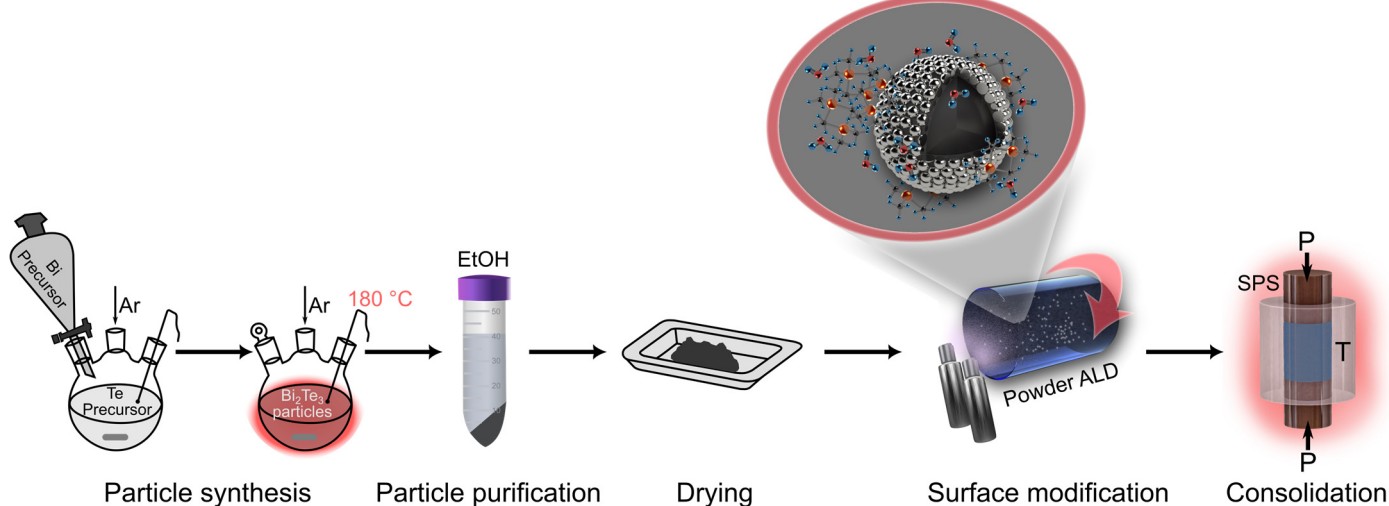

**Figure 2.** Flow chart for sample preparation of $Bi_2Te_3$ particles via solvothermal synthesis, surface modification via powder ALD and consolidation via SPS. Reproduced with permission [12]. Copyright 2022, WILEY-VCH Verlag GmbH & Co. KGaA, Weinheim.

## 3. Results and Discussion

Figure 1a shows XRR data of the ALD coating of 300 cycles $SbO_x$ using $SbCl_5$ and Sb-Ethoxide as precursors at different deposition temperatures. At a deposition temperature of 100 °C, linear growth with a growth per cycle (GPC) of 0.048 nm/cycle can be observed (Figure 1b). The GPC decreases with increasing deposition temperature in the range of 80–120 °C, with the highest GPC of 0.13 nm/cycle occurring at 80 °C (Figure 1c). The decrease in growth rate per cycle can be explained by the decomposition of the precursor at higher temperatures and the desorption of molecules from the surface. All the as-deposited films were amorphous, regardless of the deposition temperature. To study the effect of post-deposition annealing on the $SbO_x$ crystalline structure, the 20 nm $SbO_x$ film deposited at 100 °C was annealed at 350 °C for 20 min with a ramping rate of 10 C/min. The GIXRD results collected using different grazing angles show that the film was thermally stable, and no crystallization occurred during the annealing process (cf. Figure S2 in Supplementary Materials).

The left column of Figure 3 shows SEM images of the as-synthesized $Bi_2Te_3$ particles and the ALD-coated particles. The as-synthesized $Bi_2Te_3$ particles show a hexagonal shape and smooth surfaces with a thickness of about ~10 nm, which was measured using the SEM's internal measurement tools. Coating the particles with $SbO_x$ via ALD changes their surface decisively. A granular structure forms around the particles with a tendency to the formation of agglomerations, while the hexagonal shape of the particles remains and is still visible. Figure 4a shows the XRD pattern of uncoated and coated $Bi_2Te_3$ particles. The patterns can be assigned to a $Bi_2Te_3$ phase (PDF#01-086-7486) without any secondary phase, which shows the amorphous character of the $SbO_x$ coating. The thin-film GI-XRD results (Figure S2, Supplementary Materials) confirm this on the basis of the lack of reflexes of the as-deposited samples and the samples annealed at 350 °C. In the right column of Figure 3, the cross-section SEM images of the sintered pellets show that the alignment of the particles is perpendicular to the pressing direction, which is indicated by arrows. This effect is also visible in the XRD patterns of the pellets (cf. Figure 4b), which reveal a c-axis orientation (with pronounced (006) and (0015) reflexes) of the $Bi_2Te_3$ phase. The ALD coating of $SbO_x$ does not influence the preferential orientation of the $Bi_2Te_3$ particles

after pressing, but a slight shift towards higher angles of the (0015) reflex can be observed with increasing numbers of cycles (cf. inset in Figure 4b). This could be attributed to a substitutional doping effect of Sb on Bi lattice places, resulting in a relaxation of the lattice and a decreased lattice constant.

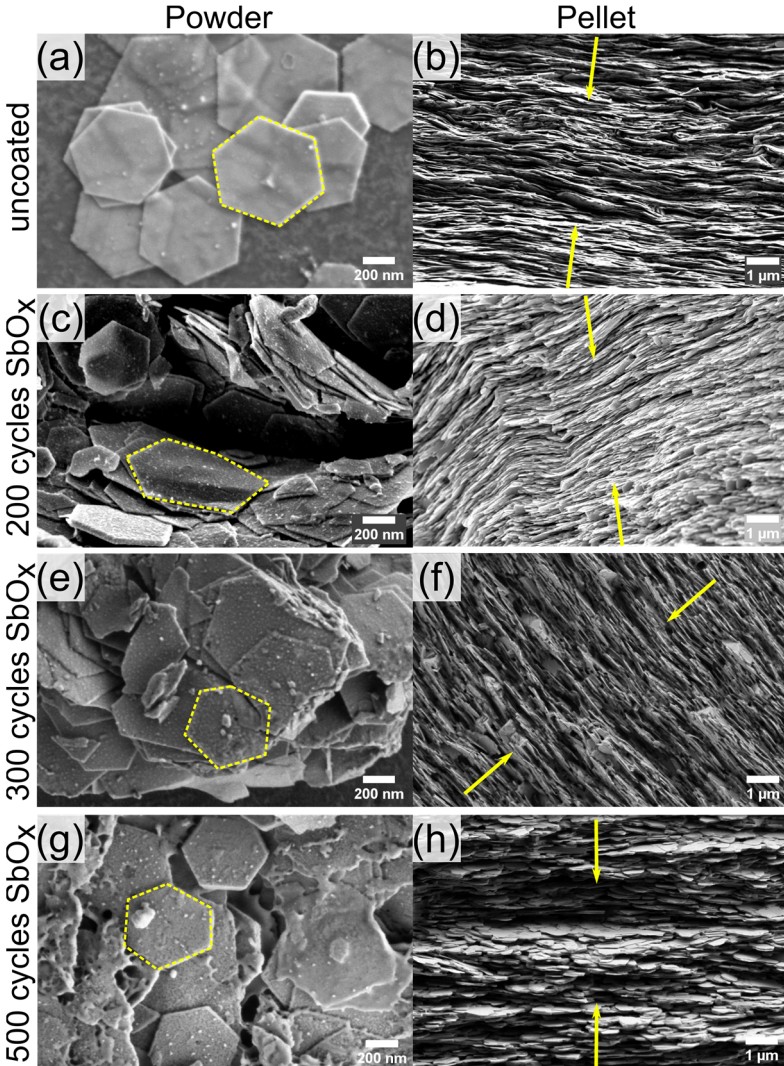

**Figure 3.** SEM images of (**a**) solvothermally synthesized $Bi_2Te_3$ particles, (**b**) cross-section of sintered pellet, (**c**) $Bi_2Te_3$ particles $SbO_x$-coated for 200 cycles and (**d**) cross-section of sintered pellet, (**e**) $Bi_2Te_3$ particles $SbO_x$-coated for 300 cycles and (**f**) cross-section of sintered pellet, (**g**) $Bi_2Te_3$ particles $SbO_x$-coated for 500 cycles and (**h**) cross-section of sintered pellet. The arrows indicate the pressing direction.

Figure 5a shows the XPS survey spectra of $Bi_2Te_3$ particles uncoated and $SbO_x$-coated with 200, 300, and 500 ALD cycles. The Bi and Te peaks resulting from the substrate powder were identified. The detailed spectra in Figure 4b–e show that the O 1s overlaps with the Sb 3d region. The uncoated sample (Figure 4b) shows just the O 1s peak, whereas the coated samples (Figure 4c–e) show Sb $3d_{3/2}$ and $3d_{5/2}$ peaks, which makes a deconvolution necessary. With increasing numbers of cycles, the $SbO_x$ species tend towards a higher oxidation state, starting with $Sb_2O_3$ at 200 cycles, a mixture of $Sb_2O_3$ and $Sb_2O_5$ at 300 cycles, and $Sb_2O_5$ at 500 cycles. In addition to the non-structural binding states of O to Carbon and Hydroxide, bindings to Bi, Te, and Sb were also detected. Additionally, the $SbO_x$-coated sample with 300 cycles showed another complex bonding of oxygen to metals, which can be attributed to a formation of Sb- or Bi-oxy-telluride.

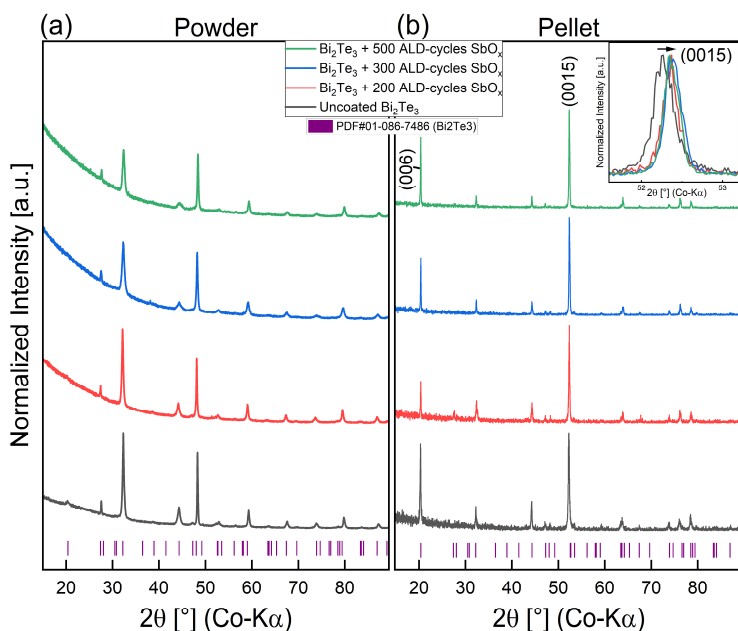

**Figure 4.** XRD pattern of (**a**) uncoated and SbO$_x$-coated powders and (**b**) the sintered pellets. All diffractograms are matched to the Bi$_2$Te$_3$ phase (PDF#01-086-7486).

A slight shift in the peaks towards higher binding energies from 529.2 eV (O1s) for the uncoated powder to 530.3 eV (Sb 3d$_{5/2}$ + O 1s) for the powder SbO$_x$-coated for 500 ALD cycles can be recognized. This shift indicates that the Fermi levels move closer to the conduction band with increasing SbO$_x$ layer thickness, which means higher n-doping of the material [22].

The specific resistivity as a function of the temperature of sintered pellets is shown in a double logarithmic plot in Figure 6. With increasing number of cycles, the resistivity decreases by one order of magnitude for 500 cycles of SbO$_x$. Since SbO$_x$ is a dielectric, the increase in conductivity is unusual, and can be explained by the formation of defects during the ALD process. The formation of Bi-O-Te and Sb-O-Te species was shown by XPS, resulting in an off-stoichiometric substrate (Bi$_{2-x}$Te$_{3-y}$) material. Independent of the sample, the resistivity increases towards higher temperatures, which could be a result of the scattering of charge carriers at the grain boundaries [2].

The Seebeck coefficient (Table 1) is negative for all samples, indicating an n-type thermoelectric material. When coated with SbO$_x$ the Seebeck coefficient can be improved by around 19% for the sample coated with SbO$_x$ for 500 cycles. A similar improvement, but with much lower layer thickness, was observed for Al$_2$O$_3$, TiO$_2$ and ZnO in a comparable study by He et al., where the coating was applied on Bismuth powder [12]. The total thermal conductivity $\kappa_{tot}$ shows a decrease from 2.4 to 1.8 Wm$^{-1}$K$^{-1}$, resulting in a $zT$ value of 0.12 at room temperature (300 K) for the sample coated with SbO$_x$ for 500 cycles. A drop in total thermal conductivity of SbO$_x$-coated Bismuth powder was observed in another work by He et al., but with a number of cycles ranging only up to 40, the reduction was minor [13]. It can be stated that the suppression of the total thermal conductivity is higher with increasing layer thickness. Compared to samples that were synthesized via ball milling, the $zT$ values are low [23]. This can be attributed to the presence of organic contaminants introduced during the solvothermal synthesis of the Bi$_2$Te$_3$ particles. Similar behavior was observed by Pettes et al., who stated that amorphous carbon can dope the surface of the nanoparticles, resulting in increased electrical conductivity, and therefore decreased $zT$ value [24].

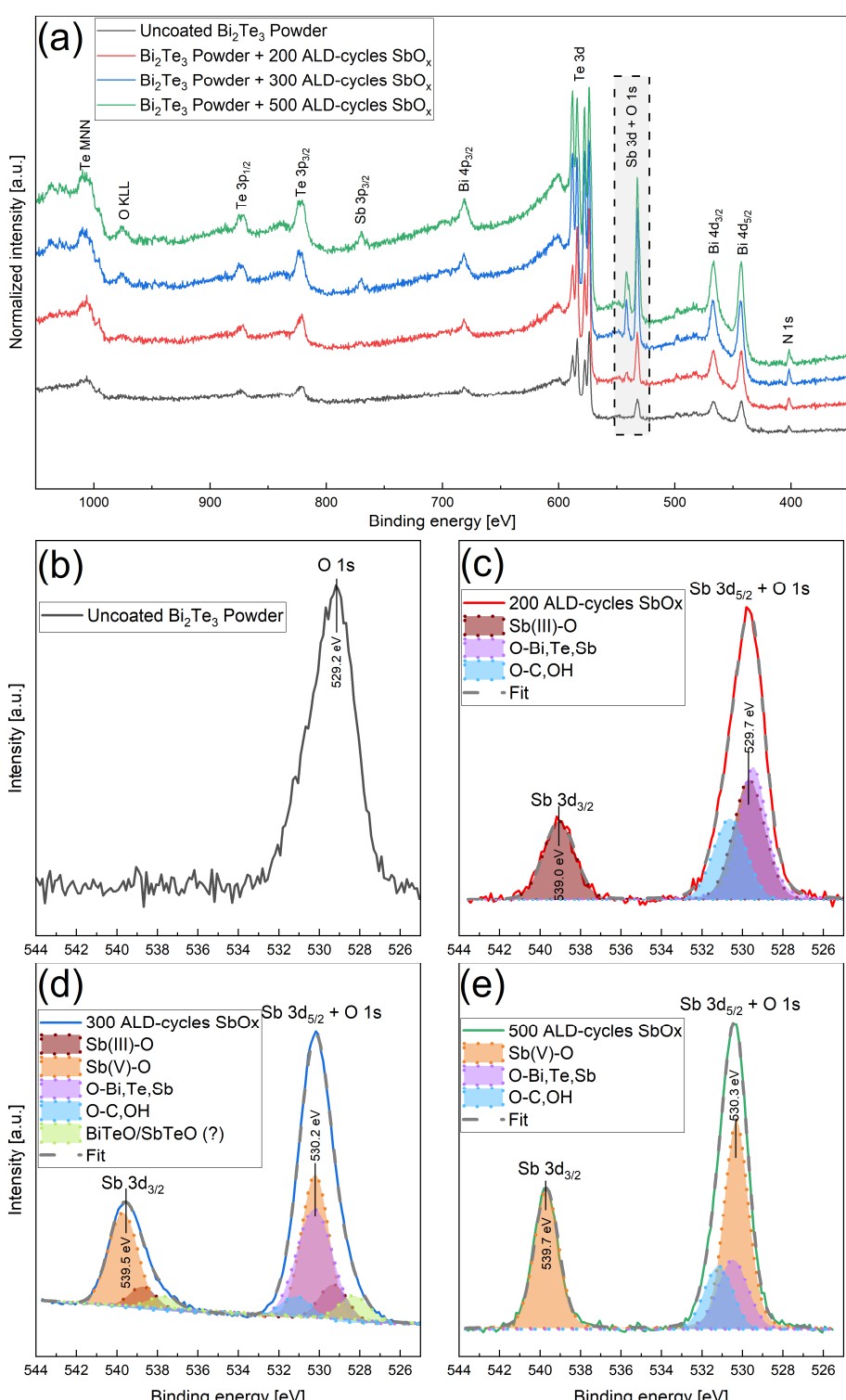

**Figure 5.** (**a**) XPS survey spectrum of Bi$_2$Te$_3$ particles uncoated and SbO$_x$-coated for 200, 300, and 500 ALD cycles. (**b–e**) Sb 3d + O 1s regions of Bi$_2$Te$_3$ particles uncoated and SbO$_x$-coated for 200, 300, and 500 ALD cycles.

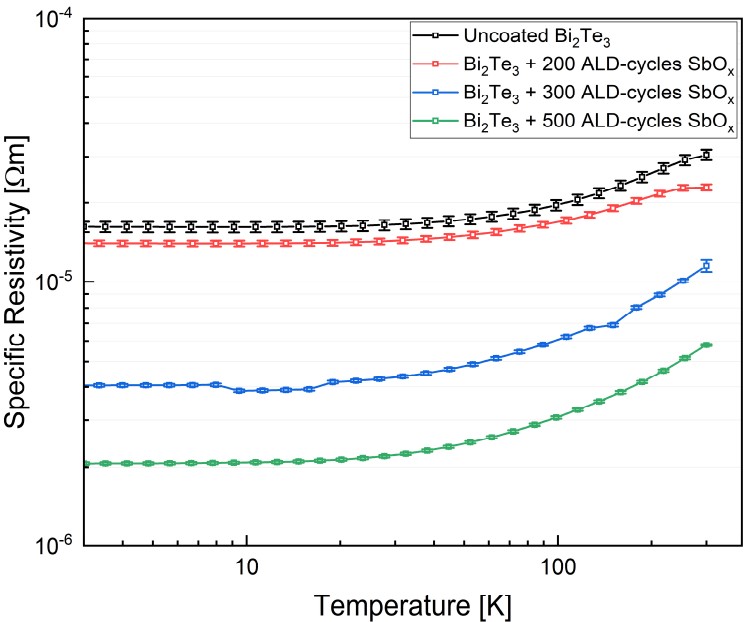

**Figure 6.** Double logarithmic plot of specific resistivity versus temperature of sintered pellets.

**Table 1.** Seebeck coefficient $S$, total thermal conductivity $\kappa_{tot}$, electrical conductivity $\sigma$ and calculated $zT$ value at room temperature (300 K) for uncoated and $SbO_x$-coated sintered samples.

| Sample | Seebeck Coefficient $S$ | Thermal Conductivity $\kappa_{tot}$ | Electrical Conductivity $\sigma$ | $zT$ |
|---|---|---|---|---|
| | [$\mu VK^{-1}$] | [$Wm^{-1}K^{-1}$] | [S/m] | |
| Uncoated | −52.6 | 2.4 | $3.28 \times 10^4$ | 0.01 |
| 200 cycles $SbO_x$ | −59.7 | 2.0 | $4.35 \times 10^4$ | 0.02 |
| 300 cycles $SbO_x$ | −62.5 | 1.8 | $8.70 \times 10^4$ | 0.06 |
| 500 cycles $SbO_x$ | −64.8 | 1.8 | $1.73 \times 10^5$ | 0.12 |

## 4. Conclusions

A new water-free $SbO_x$ ALD recipe was developed using $SbCl_5$ and Sb-Ethoxide as precursors that could be used for the coating of materials that are sensitive to moisture, since the recipe does not involve the use of a conventional oxidizer like water or $H_2O_2$. As an example, powder samples of the well-known thermoelectric material $Bi_2Te_3$ were coated and the thermoelectric properties of spark plasma-sintered pellets as well as the structural influence of the coating were studied. With the amorphous $SbO_x$ coating, a shift in the oxidation state from +III to +V was demonstrated with increasing cycle number and layer thickness. The specific electrical resistivity decreased with increasing layer thickness by one order of magnitude, which was attributed to the doping effect of Sb into the $Bi_2Te_3$-lattice and excess carbon from the solvothermal synthesis. The doping effect was confirmed by a shift in the (0015) reflex in the XRD pattern towards higher angles and a shift in the Sb $3d_{5/2}$ + O 1s peak towards higher binding energies with increasing $SbO_x$ coating. The Seebeck coefficient improved by about 19% owing to the dielectric properties of the coating, while the total thermal conductivity decreased by about 25%. This results in an overall increase in the $zT$ with $SbO_x$ content. These findings show that powder ALD is a very effective tool for manipulating the electric and thermoelectric properties of materials via surface and interface engineering.

**Supplementary Materials:** The following supporting information can be downloaded at: https://www.mdpi.com/article/10.3390/coatings13030641/s1. Figure S1: AFM results of (a) 700 cycles ALD-SbOx and (b) 500 cycles deposited at 100 °C. The layer thickness was determined measuring a step in line

scan mode; Figure S2: Gracing incidence XRD results of as-deposited and at 350 °C annealed samples. The coating is amorphous.

**Author Contributions:** Conceptualization, S.L.; methodology, F.M., S.H., J.Y. and M.H.; validation, S.L., A.B. and S.H.; formal analysis, S.L., J.Y. and M.H.; resources, K.N.; data curation, S.L., S.H. and J.Y.; writing—original draft preparation, S.L.; writing—review and editing, A.B.; supervision, K.N. and A.B.; project administration, K.N.; funding acquisition, K.N. All authors have read and agreed to the published version of the manuscript.

**Funding:** This work was supported by the European Union's Horizon 2020 research and innovation program under grant agreement No 958174.

**Institutional Review Board Statement:** Not applicable.

**Informed Consent Statement:** Not applicable.

**Data Availability Statement:** The data presented in this study are available on request from the corresponding author.

**Acknowledgments:** Special thanks to Ronald Uhlemann for the preparation of the schematic illustrations.

**Conflicts of Interest:** The authors declare no conflict of interest.

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
