# Peer review of "Water-Free SbOx ALD Process for Coating Bi2Te3 Particles"

_coatings, doi:10.3390/coatings13030641_

Round 1

Reviewer 1 Report

The authors developed a water-free Atomic layer deposition (ALD) process to homogeneously deposit SbOx using SbCl5 and Sb-Ethoxide as precursors The coating was applied on Bi2Te3 particles. The authors used advanced experimental techniques to characterize the coating

THE WHOLE WORK IS INTERESTING.

POINTS FOR IMPROVEMENT:

1. Please make a patent literature review for similar methods & materials.

2. What are the limitations of your method?

3. Could this process applied to industrial scale?

4. Please, provide a Table with the technical characteristics of available commercial coatings.

5. Please, compare the prepared coatings properties with the properties of available coatings

Author Response

The authors developed a water-free Atomic layer deposition (ALD) process to homogeneously deposit SbOx using SbCl5 and Sb-Ethoxide as precursors The coating was applied on Bi2Te3 particles. The authors used advanced experimental techniques to characterize the coating

THE WHOLE WORK IS INTERESTING.

POINTS FOR IMPROVEMENT:

Q1. Please make a patent literature review for similar methods & materials.

R1. Thanks for your suggestion. Indeed, in 2021 we published a comprehensive review paper on the topic “Interface/Surface Modification of Thermoelectric Materials”. This review paper is cited as a reference [2] in the manuscript. For the interest of the reviewer and readers, we added some relevant parts to the introduction, highlighted in red color.

Q2. What are the limitations of your method?

R2. The time of the process can be considered a limiting factor due to the self-limiting character of ALD reactions. Also, for the substrates with a high surface area, sufficient time has to be considered until all surfaces are saturated with chemicals. However, in surface or interface modifications of powders, a few cycles already show changes in electrical properties which makes the process short.

Q3. Could this process applied to industrial scale?

R3. Yes, ALD coating of higher amounts of particles is already industrially available (e.g. companies like Forge Nano provide devices for particle coatings for industry.) We added the sentence into the manuscript: “Upscaling of this coating procedure to industrial scale is easily possible using a larger rotating drum or utilizing a so-called fluidized bed reactor (FBR). In regards to the larger surface area, the exposure time should then be increased.”

Q4. Please, provide a Table with the technical characteristics of available commercial coatings.

The authors appreciate your suggestion, however, such a table would go beyond the scope of this study. Available commercial coatings range from pigments/paints, and finishing of decorative surfaces to performance improvement of battery materials and protective coatings against moisture/corrosion etc.

Q5. Please, compare the prepared coatings properties with the properties of available coatings.

R5. Thank you for the suggestion; we compared the deposited coating characteristics in this study with those in the literature, highlighted in red.

Reviewer 2 Report

Very interesting subject.

The used precursors applied as coating of materials are sensitive to absorbe moisture ?  What kind of applications can you developed with these type of structures?

What is the value added of your work to scientific community? 

is this process considered as "green" technology? 

Author Response

Q1. The used precursors applied as coating of materials are sensitive to absorb moisture?

R1.The precursors itself/alone can react with moisture to form e.g. oxides. Letting them react with each other in a controlled way in an ALD chamber leads to the controlled reaction to form the SbOx which is a very stable coating. Please note that all the chemical precursors are handled and stored in the glove box and extra attention is paid to avoid any exposure to the air during the filling and installation of precursor canisters to the ALD devices.

Q2. What kind of applications can you developed with these type of structures?

R2. The sintered and compacted particles can be used to form thermoelectric modules (so-called legs). A leg-pair (n- and p-type) is a thermoelectric module. We added this information to the manuscript, highlighted in red color.

Q3. What is the value added of your work to scientific community?

R3. The main point added to the scientific (ALD) community is the newly developed water-free SbOx-recipe and that surface and interface modifications are doable by applying the technique of Powder ALD. We believe that this is sufficiently clarified in the manuscript.

Q4. Is this process considered as "green" technology?

R5. Application of Thermoelectric materials in the conversion of waste heat to electricity can be, in general, considered as green technology. However synthesizing and modification of them require materials which sometimes are rare or toxic like Te, Se, or Sb. Therefore, the fabrication and synthesis of thermoelectric materials cannot be considered as a green process.

Reviewer 3 Report

The authors reported the manuscript entitled "Water-free SbOx-ALD-process for coating Bi2Te3-particles" to enhance the thermoelectric properties via interface engineering. The experimental results show that surface engineering via Powder ALD is an effective tool to improve critical properties of thermoelectric materials like electrical conductivity and the Seebeck coefficient. Although the Paper  has well written some points should be cleared before acceptance  for publication:

1. The GPC decreases with increasing deposition temperature in the range of 80 – 120 °C showing the highest GPC at 80 °C of 0.13 nm/cycle. Why?

2. WhyaAfter coating the particles with SbOx via ALD, the surface decisively changes?  Explain it.

3. Compare your results with already published data.

4. Novelty and objectives should be clearly mentioned.

Author Response

The authors reported the manuscript entitled "Water-free SbOx-ALD-process for coating Bi2Te3-particles" to enhance the thermoelectric properties via interface engineering. The experimental results show that surface engineering via Powder ALD is an effective tool to improve critical properties of thermoelectric materials like electrical conductivity and the Seebeck coefficient. Although the Paper has well written some points should be cleared before acceptance for publication:

Q1. The GPC decreases with increasing deposition temperature in the range of 80 – 120 °C showing the highest GPC at 80 °C of 0.13 nm/cycle. Why?

R1. Thanks for the remark. Two main possibilities can be considered here: decomposition of the precursors and desorption of molecules from the surface, as also was observed in the literature. We added this to the manuscript, highlighted in red color

Q2. Why after coating the particles with SbOx via ALD, the surface decisively changes? Explain it.

R2. The sentence in the manuscript was unartfully expressed. We wanted to express that we see a smooth uncoated surface and after coating a decisive change in the morphology. The sentence was rewritten as: “Coating of the particles with SbOx via ALD changes the surface decisively. A granular structure forms around the particles with a tendency to the formation of agglomerations while the hexagonal shape of the particles is remaining and still visible.”

Q3. Compare your results with already published data.

R3. We compared our results to two similar studies where the thermoelectric material Bismuth was coated with different oxides via powder ALD and added this comparison to the results and discussion section, highlighted in red color

Q4. Novelty and objectives should be clearly mentioned.

R4. Thanks for your suggestion. The novelty is the water-free SbOx-ALD-process which was reported here for the first time. This is mentioned in the Abstract and the Conclusion in the revised version of the manuscript. One main objective of this study was the interface and surface modification via Powder ALD which is for our understanding emphasized enough in the revised version of the paper.

Round 2

Reviewer 1 Report

INTERESTING WORK.